# Intimacy and Darkness: Feminist Sensibility in (Post)socialist Art

Jana Kukaine 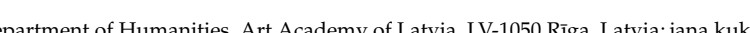

Department of Humanities, Art Academy of Latvia, LV-1050 Rīga, Latvia; jana.kukaine@gmail.com

**Abstract:** This article assembles feminist articulations scattered across art histories and theories of Eastern and Central Europe, in order to reveal their potential, not only for foregrounding postsocialist feminist perspectives, but also for enriching the vocabulary and expanding temporal geographies of transnational feminist debates. By attending to intuitive, latent, reluctant, proto-, para-, unofficial and soft feminisms, this article establishes a peculiar feminist sensibility that is attuned to Central and Eastern European women artists' approaches to everyday, embodied and affective experiences via the critical endorsement of intimacy and darkness.

**Keywords:** postsocialist feminisms; feminist aesthetics; feminist art in Eastern and Central Europe; everyday life; close to one's skin; viscerality

In 2005, Alice Červinkova and Kateřina Šaldova conducted research on the relationship of women artists towards feminism in the Czech Republic, featuring an anonymous confession of one interviewee, perhaps a woman artist: "I use that word feminism only intimately and when it is dark". These words were later quoted by Bojana Pejić in the *Gender Check: A Reader. Art and Theory in Eastern Europe*, that is the publication which until now has provided the most comprehensive overview of the feminist perspectives in the arts in the region (Pejić 2010b, p. 34). Nonetheless, this seemingly random and even amusing utterance, not only describes how somebody might feel about feminism but, as I would like to suggest in this article, it also offers a significant contribution to the conceptual grounding of postsocialist feminism via a theoretical rethinking of intimacy and darkness whose usage may extend far beyond postsocialism.

However, any references to postsocialist feminism, and especially writing *as* a postsocialist feminist frequently—at least implicitly—demands a disclaimer about the politics of location and the cultural situatedness of this feminist stance, both *within* and *outside* the postsocialist situation. Postsocialism as a term should, neither be interpreted exclusively in a temporal manner (i.e., as a period after socialism, or the end of the Cold War), nor is it simply a part of the local history of a particular region, but rather it is a global human condition that invites revising socialist heritage from a transnational point of view (Koobak et al. 2021, pp. 1–3). This article will follow the same logic and argue that expanding the limits of postsocialism is theoretically and politically rewarding, where this adaptation of a transnational point of view entails attentiveness to vernacular cultures and considers the significance of local nuances. First, I will outline the intellectual trajectory of postsocialist feminism and hint at its ambiguous relations with other established transnational feminist theories. Then, I will search for "feminisms of their own"[1], putting forward a fragmented map that represents "islands of interest in feminisms"—a metaphor used by Slovak art scholar Jana Geržova (2010, p. 309)—in late socialist and postsocialist art theories in Central and Eastern Europe, and from my own perspective living and working in Latvia. These "other feminisms" will demonstrate that intimacy and darkness can be used as aesthetic categories of analysis that attest to a peculiar and often ambiguous feminist sensibility typical of the region's culture and art, and would anchor feminism in everyday life, as well as lived and affective experiences. Thus, thinking through feminist sensibility

in (post)socialist art opens up new perspectives for debates in feminist aesthetics and art research, as well as equipping postsocialist feminist scholars with critical vocabulary and tools for further inquiry.

### The Uncanny Close Other

The challenges of installing postsocialist feminism and claiming its space in transnational feminist agendas, as well as negotiating the inequality prevailing in global feminist debates (including the burden of translation) have already been noticed and explored.[2] Tlostanova's ironic question "if the post-Soviet can think?" (Tlostanova 2015)[3] leads to the next assumption about postsocialist feminism as the "missing other" in transnational feminist debates (Tlostanova et al. 2019). While there are several areas of interest and motives that tend to organize transnational feminist theorizing, often adapting internalized narratives of progress (Koobak and Marling 2014), the North-American women's art movement continues to possess a definitional monopoly and functions "as a yardstick with which to measure developments everywhere else" (Hock 2018, p. 14). Contemporary feminist art research often revolves around the "Anglo-American axis" and addresses Western Europe as an extension of white America's cultural "home", as well as an unmarked normative category (Meskimmon 2007). While I do not intend to homogenize or universalize all feminist theories that have stemmed from or hover around this axis, its presence is especially vivid in the attempts to advance a postsocialist feminist art history and theory in Central and Eastern Europe. Patterns of epistemic inequality, as well as voluntary intellectual self-colonization which is heightened and normalized by the prevailing Eurocentric politics of the region, and the desire to "get rid of" its socialist past[4], are just a few symptoms.

As a cultural phenomenon and intellectual trajectory, postsocialist feminism emerged in the 1990s when the fall of the Soviet empire opened up new terrains for (predominantly Western) feminist scholars, including in feminist art history, critique, and aesthetics. However, soon after the "discovery" of Eastern Europe, several disappointing episodes followed, epitomized by the advice for the "Easterners" to take a few intensive courses in contemporary feminist theory, in order to "catch up" with the West, as Edit Andras had observed it. By ignoring the differences in context and the specificity of Eastern Europe, the region gradually slipped into the category of "Other" (Andras 1999, p. 8), while "local" postsocialist feminist scholars have been assigned the role of the dutiful daughters whose distant stepmothers often did not treat them as intellectual equals (Marling 2021, p. 94).

The following decades strengthened the perception of postsocialist feminism as a messy and vague "copy of the West", an uncanny combination of differences and similarities. The uncanny otherness of postsocialist feminism manifests in being "similar to the West but not similar enough, while also registering as different but somehow *not different enough*." (Tlostanova et al. 2019, p. 81). The interplay of strangeness and familiarity, sameness, and difference, as well as distance and intimacy, the known and the unknowable, enables the framing of postsocialist feminism as a Close Other—the term used by art theoreticians Bojana Pejić (1999), Piotr Piotrowski (2014), and Beáta Hock (2018) to describe the relations of Eastern Europe to the West. This relation is also influenced by the binary pattern of copy and original—not because the West is viewed as the authentic and homogenous producer of true feminism, but because postsocialist feminist debates often orient themselves in accordance with theories introduced by mostly anglophone publishers, academic journals, etc. On the one hand, the Eurocentric orientation of contemporary debates on gender equality, including topics, such as violence against women and sexual harassment, the double burden, work-life balance, as well as the #*metoo* movement, is perceived by many as mimicking the Western feminist discourses which have now become "fashionable" in Central and Eastern Europe under neoliberalism, while the accomplishments of socialist feminists are either overlooked as too ideological or discarded as rather worthless remnants of the Soviet past. On the other hand, the impetus of overcoming the Soviet past has been strengthened by multiple forces, e.g., the alleged "fatigue" or even "allergy" of "women's emancipation" associated with the former Communist regime, the revival of traditional

gender roles, the paradoxical fusion of neoliberalism and nationalism, as well as the popular rhetoric of "catching up with the West", etc. (Einhorn 1993; Smith 1996; Pachmanová 2010; Traumane 2012; Deepwell and Jakubowska 2017; Marling and Koobak 2017; Marling 2021). Socialist feminist herstories, as well as feminist insights of earlier periods only recently have started to gain visibility, although—at least in Latvia—this research is carried out sporadically and in precarious material conditions.

In the concept of a Close Other, closeness is both uncomfortable and surprising and reminiscent of the initial interpretation of the uncanny in psychoanalysis, as a disturbing fallout of the return of the repressed. According to Alexandra Kokoli, the uncanny can also be used as a feminist ally "in its attempt to forge subversive countercultural strategies, to claim a place in the canons of creative practice and critical theory, and to revolutionize them in doing so" (Kokoli 2016, p. 3). To conceive of Eastern European feminist discourses as "uncanny", designates not only its homelessness, but also its unhomeliness, i.e., the ability to initiate estrangement and confusion, while its ambiguity derives from being "neither in nor out" but rather "in-between", as suggested by theoretician Martina Pachmanová in her pioneering essay on the subject (Pachmanová 2010, p. 37). Attributes, such as elusiveness, theoretical instability, as well as conceptual obfuscation may also offer some advantages for rethinking postsocialist feminism in an affirmative way and embracing its potential for the enrichment of critical vocabularies, challenging canons, and maybe even revolutionizing transnational feminist research. Moreover, if attending to the idiosyncratic feminist terminology of Central and Eastern European art discourses, these potentialities are especially pronounced, as I will attempt to demonstrate in what follows.

**The Islands of Interest in Feminism**

Feminist research in Eastern and Central Europe is abundant in challenges, including facing the banalization and misinterpretation of key Western feminist ideas and concepts, as well as dealing with the ghettoization of feminist endeavors or attempts to marginalize, overlook or neutralize them, e.g., by relabeling them as "feminine" or "women-friendly", etc. (Dimitrakaki 2005, p. 271; Pachmanová 2010, p. 41). The common attitudes to feminism in Central and Eastern Europe—denial, doubt, stereotypical ideas, a reluctance to take it seriously, and an overall anxiety about feminism have been addressed by multiple researchers (Einhorn 1993; Andras 1999; Baigell and Baigell 2001; Haan et al. 2006; Pachmanová 2010, 2019; Kivimaa 2012; Pejić 2010a; Hock 2018), and specifically with regard to Latvia (Traumane 2012; Kukaine 2021). Yet, despite the difficulties, several art historians and theorists, as well as artists, have recognized or enhanced feminist dimensions or "sensibilities", in the art of Central and Eastern Europe, introducing an idiosyncratic and innovative terminology: focused on "intuitive", "latent", "reluctant", "proto-" and "para-feminist" approaches, as well as calling them "unofficial" and "soft". In their original contexts, these concepts are often used as epithets or mentioned casually as descriptions, without providing any in-depth explanation or theoretical grounding. However, when these feminist islands are assembled together, a critical vocabulary emerges, that, on the one hand, expresses the conflicting feelings towards feminism prevailing in the region, and, on the other hand, marks local differences, based on the interplay of the notions of intimacy and darkness. I will shortly trace their discursive origins and usage.[5]

The first and perhaps the most "traditional" concept is "intuitive feminism". This term has been used by the Croatian art scholar Ljiljana Kolešnik to analyze the works of the performance artist Vlasta Delimar. In her works *Fuck Me* (1981), *Visual Orgasm* (1981), *I Love Dick* (1982), etc., the artist addresses topics, such as violence against women, motherhood, aging, and sexuality, especially highlighting patterns of intimacy, female sexual pleasure, and relationships with men. Many of Delimar's works were considered provocative and even scandalous; the artist was accused of pornographic content, narcissism, and even misogyny (Tumbas 2018), which testifies to the uncanny dimension of Delimar's artistic insights. However, like many artists in the region, Delimar is unwilling to associate her work with feminism, therefore Ljiljana Kolešnik framed her art practice as "intuitive

feminism" ([Kolešnik 1997](#)). While conventionally, intuition evokes or is seen as an attribute of "irrational femininity", and to use this term may seem to be a return to traditional binary oppositions, in recent feminist thinking, intuition is now viewed as a strategy to depart from logocentric patriarchy, in order to embrace embodied and affective ways of knowing. Likewise, in Delimar's art, "intuitive feminism" stands for thinking "with or through the skin" ([Ahmed and Stacey 2001](#), p. 1), intimately unfolding a feminist sensibility by reflecting on bodily and affective everyday experiences. It should be likewise noted that in the art practice of Delimar (as well as with many other feminist artists), a feminist agenda is almost indiscernible from a desire to resist and defy censorship and, by rewriting the scripts of acceptable female identity and "decent" behavior, her works challenge not only patriarchal social norms, but also an authoritarian State's power and its ideology. "Intuitive feminism", therefore, is idiosyncratic—it is closed, peculiar, and theoretically unbounded by established patterns of feminist inquiry. It offers a personal gesture of protest and not a collective revolt, and being untraceable and nomadic, it resists unhomeliness by finding a home in one's skin.

Another island of feminist interest is "reluctant feminism". Art historian, Beáta Hock, applies this notion when referring to the authors who research women's art in Eastern Europe (not only in the post-1945 period, but also in the decades following 1989) and are intrigued by the "attitude in which "feminist overtones" mix with an indifference toward discourses of "Western second wave feminism" ([Hock 2018](#), p. 14). A telling example of this intrigue are the famous words of the Polish artist Natalia LL, when she expresses an irritation with "Western feminisms" by asserting that Polish women in 1970 already had everything their Western comrades were fighting for, namely, the hardships of maternity and the right to suffering, hard work, and superhuman responsibility (cited in [Bryzgel 2018a](#), p. 166). While the artist's words are definitely ironic, her irony also seems to suggest that feminist dimensions in art, aesthetically or conceptually, are not always identical or similar to previously established or expected Western standards. Likewise, it is highly credible that the Western feminist discourses neither spoke on behalf of Eastern European women nor fully reflected the reality of their lives ([Hock 2018](#), p. 14). However, it does not necessarily follow that Eastern European artists were eager to embrace another, non-Western feminist agenda (e.g., feminisms found in many so-called developing countries, or the Third World). The "reluctant feminist" sensibility expresses this attitude that lingers between irritation and ignorance, interest, and denial. Feminism seems to be put "on hold", postponing the decision of affiliation, to win time for a new and more rewarding identity that is based on their cultural and historical reality of what was lost in Communist versions of equality or optimistically proposed by neoliberal feminism. Like "intuitive feminism", the "reluctant feminist" however seeks to avoid what is known and visible and embraces more obscure and opaque ways of being.

On the postsocialist feminist map, there are also islands of feminisms that are not primarily derived from the intentions or preferences of the artists, but rather originate as a response to social and cultural processes: "latent feminism" is one of these. The term is featured in the writings of Slovak art theorist Zora Rusinová, for example, when she is interpreting "the hidden aspects of feminism" in the works of artist Jana Želibska ([Rusinová 2010](#), p. 145). According to Rusinová, it is possible to detect how particular works of hers, "albeit more or less intuitively and often through metaphors and symbols [ . . . ] featured a codified focus on the critical principles of feminist art, aimed at confronting the "manly privileged subject" in the process of viewing the world—where the womanly was only a passive object exposed to a controlling gaze—with the womanly active subject in self-reflection" (*ibid*.). "Latent feminism" describes both Želibska's artistic environment and insights into the contexts of her oeuvre, based on comparisons with Western Europe: so, while Western feminist art gained visibility and vigor, fueled by the women's rights movement, political activism, and feminist accomplishments in academia, the conservative Slovak art scene was controlled by communist ideology and standards of socialist realism were imposed. Only after 1989 did Želibska's feminist ideas come "fully to

light" (Rusinová 2010, p. 149) as before that, the interest in feminism had to be hidden or subdued. Today, it appears relatively easy to recognize a feminist orientation in Želibska's art, due to its critical interest in gender roles, sexuality, the female body, etc., as well as a choice of materials (such as jewels and mirrors) and techniques (e.g., embroidery and sewing), and to see her feminism as existing but not yet manifested or fully articulated. The term "latent" is indicative of social and cultural awareness and a readiness for feminism, outlining a feminist potentiality for future transformations and encounters. From this point of view, it can be argued that the function of "latent feminism" is affectively corporeal—like an affect, it is beyond representation, namely, it does not fit into the paradigm of official cultural patterns, but nonetheless "unsettles us into becoming someone other than who we currently are" (Hemmings 2005, p. 549), as it comes into full light. Latent feminism initiates social change while remaining concealed, "invisible", and unwilling to manifest the usual feminist "symptoms" which are attached to the previously mentioned Anglo-American axis, while its affectivity provides possibilities to locate it in the realm of everyday life, the personal, and the intimate.

These ideas about feminism as "latent" are further developed by examining the parallel concepts of "proto-feminism" and "para-feminism". Art historian Mark Allen Svede argues that since the 1970s, "women were not particularly disadvantaged in the institutional art world" and the arrangements of the Latvian art scene with prominent painters, such as Džemma Skulme and Maija Tabaka, "undermined the key Western feminist art premises" regarding institutional/art scene discrimination against women (Svede 2002, pp. 242–43). Although the discussion about how to measure the feminist temperature of late Socialism and whether and in what way it undermined key Western premises is still ongoing, there are many considerations that cast significant doubt on these Soviet models of gender equality. For now, I am not going to delve into the debate. What interests me is how Svede argues that "proto-feminist" sensibility can be detected in, for example, the Latvian artist Ilze Zemzare's work in the 1960s. This painter is virtually unrecognized to this day and marginalized for several overlapping reasons: that she has been "eclipsed by her husband's professional stature, pigeonholed by her training as a decorative artist, [and] hindered by her own modesty" (Svede 2002, p. 241), and was also traumatized and silenced by the political repression against her partner. Zemzare's work *The Path of Life* (from 1966) depicts two vulnerable female figures and a strand of barbed wire which Svede understands as emblematic of the average woman's experience in Soviet Latvian society, and this is how he foregrounds a "proto-feminist" perspective. The critical potential of the term "proto-feminism" lies in its semantic connotations for claims to be "the first" and "the earliest", that which is "before" and that which is "giving rise to" something which becomes feminism after 1968 or in the 1970s. Thus, introducing this kind of (post)socialist and retrospective attribution of "proto-feminist" insights to transnational feminist frameworks, challenges established historiographies and narratives of progress, e.g., by inviting us to revise the relations of "before" and "after", an original and a copy, center and periphery, stimulating the process of both the provincialization of the West and instead advancing horizontal and rhizomatic perspectives, not only in writing feminist art histories, but also in thinking about contemporary feminist art. "Proto-feminism" may therefore be a conscious strategy to "escape" the prevailing feminist temporalities and avoid structures of knowledge and classification that are allegedly universal, in order to give rise to a version of a more inclusive and less hierarchical feminist knowledge.

The disruptive effect on the hierarchies of a feminist art canon is also produced by another feminist sensibility, namely, the "para-feminist". If "proto-feminism" is a temporal intervention, then "para-feminism" is a spatial one—it proposes to locate postsocialist feminist art discourses "beyond" and "further than" the Western center, thus expanding feminist geographies. The term's provenance is associated with the works of Amelia Jones (2012), however, it enters into postsocialist feminist discussions with significant alterations introduced by art scholar Alise Tīfentāle and her research on Latvian photographer Zenta Dzividzinska (aka ZDZ[6]). For Jones, "para-feminism" offers a way to

rethink the "temporal and embodied relationality" which is based on new and non-binary feminist politics—"it does not *supersede* or overthrow earlier feminist models" (Jones 2012, p. 183) but rather builds upon them, exploring and extending, pushing the boundaries of past strategies, while avoiding tendencies to binary oppositions and universalisms (McKenzie 2016, p. 123). According to Tīfentāle, in the 1960s, ZDZ was one of the few women photographers in Riga whose work was acknowledged in both local and international photography scenes, although in later decades and until recently, her artistic contribution has been largely neglected due to sociopolitical and cultural reasons (Tīfentale 2021). One of the most noteworthy ZDZ works is the series *A House Near the River.* It captures the daily life of three generations of women living in a small house in the country. Women in these images were not intended to be a source of pleasure for heterosexual male spectators, as was common in Latvian photography of the 1960s. Instead, these women appeared "as self-sufficient individuals, preoccupied with their chores and not performing for the camera" which was "the most shocking for the cohort of ZDZ's mostly male peers" (Tīfentale 2021). Tīfentāle proposes that these photographs can be seen "perhaps as para-feminism (to use Amelia Jones's term) because explicitly feminist critique did not emerge in Latvian art until the 1980s". "Para-feminism" thus indicates an implicit sensibility that is nonetheless choking and at odds with the dominant ideas about art.

A similar "para-feminist" stand can be attributed to ZDZ's series of self-portraits, created in the same decade (Figure 1). According to Tīfentāle, these images, where the artist experiments with photographic and optic means (e.g., fish-eye lenses and distorting reflections), in order to produce rather unflattering depictions of her body, not only predate the subgenre of self-portraiture known as the "selfie" but also question the assumption that women in photography have to look pretty. Thus a "para-feminist" attitude upholds the aesthetic category of unsightly, unattractive, and even ridiculous femininity, which can be viewed as part of "a broader, transnational narrative of "soft" or "quiet" resistance" (Tīfentale 2021). These adjectives, as will be demonstrated later, are rather telling and significant for postsocialist feminist sensibilities.

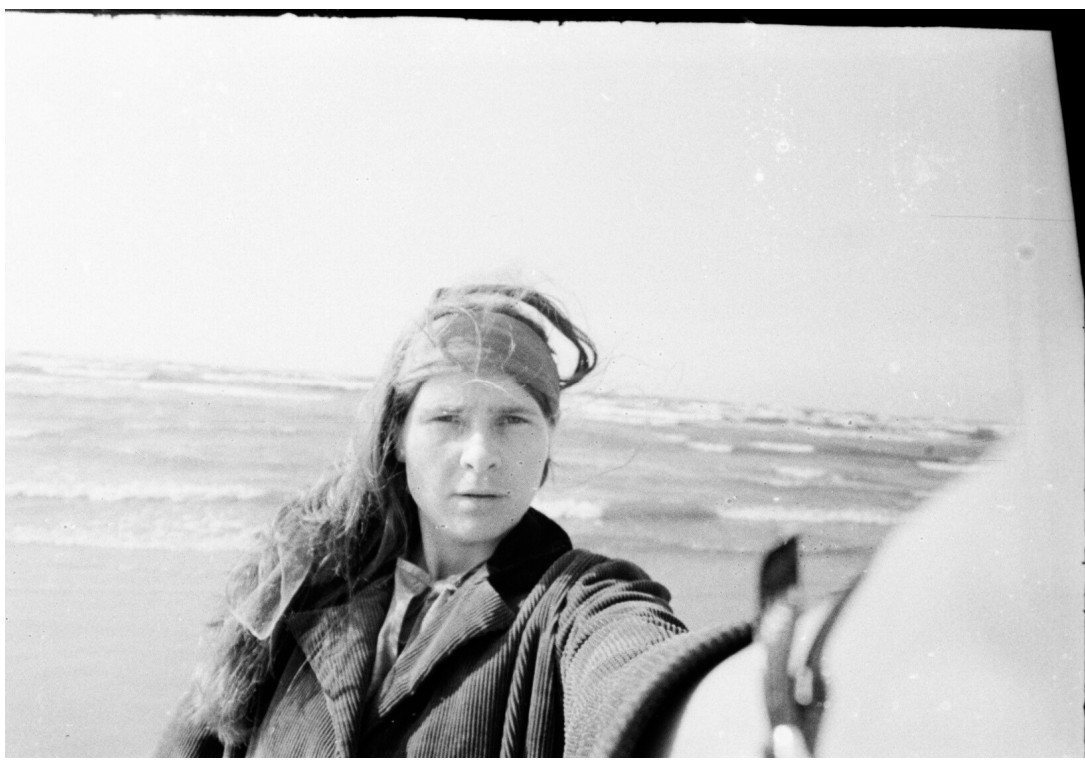

**Figure 1.** Zenta Dzividzinska, *Self-portrait*, 1966. Gelatin silver print, 10 cm × 18 cm. Courtesy Zenta Dzividzinska and Juris Tifentals Archive, www.artdays.net (accessed on 18 September 2022).



Tīfetāle's text itself can be identified as a "para-feminist" reading of ZDZ's work attempting to inscribe Zenta Dzividzinska in the feminist art tradition. Interpretating ZDZ's work as standing "side by side" or "further than" that tradition, Tīfetāle pushes its boundaries and offers a new kind of temporal and embodied relationality. This is how a postsocialist "para-feminist" sensibility can be located side by side with other earlier and more elaborated feminist traditions in the West, while encouraging critiques of spatial and temporal models that have shaped dominant feminist narratives and maintain their main axes. It is worth noting that this "para-feminist" approach to rethinking and recontextualizing ZDZ's work was carried out in 2021 through a performative exhibition curated by Zane Onckule *I Don't Remember a Thing: Entering the Elusive Estate of ZDZ* (Latvia: *Kim*? Contemporary Art Center, 2021). The exhibition brought ZDZ's only partially known photographic legacy side by side with an artist Sophie Thun and an archivist Līga Goldberga, intimately materializing a temporal and embodied kinship for tracing the hidden and unknown continuity of generations of women artists and researchers.

The final two specimens of this Eastern European feminist collection are "soft" and "unofficial" feminisms. These terms are coined by artists. In her research about performance art in Eastern Europe, Amy Bryzgel quotes the Romanian artist Lia Perjovschi, who describes herself as an "unofficial feminist", by which she means that she is working intuitively rather than in relation to feminist theory or referring to examples of feminist art (Bryzgel 2018a, p. 170). The mention of intuition as an attribute of postsocialist feminism here, is telling in itself: however, it is contrary to the previously mentioned "intuitive feminism". Here, the choice of the term does not indicate a distance from feminism as such, but rather suggests a private, idiosyncratic, and underground usage. Significantly, in postsocialist art theories, the term "unofficial" is loaded with meanings since it often is indicative of art processes that did not fit into the officially imposed patterns of art making, attempted to transgress the rules set by the communist party and implied that the work was most likely also illegal. "Unofficial feminism" is not only publicly unaccepted but also unacceptable, i.e., potentially dissident and subversive.

Bulgarian artist Adelina Popnedeleva, by contrast, portrays herself as a "soft" feminist, explaining that she is against all forms of hierarchy and that her work is about equality, rather than focusing solely on gender (Bryzgel 2018b). Popnedeleva's works refer to traditional women's duties (such as washing clothes, weaving, etc.), the naturalization of female pain, and gendered expectations (e.g., *Are You Blonde Enough to Survive*? in 2014) and are often very personal, as in *Psychotherapeutic Performance* (2004), where the artist tried to recover from migraines with the help of a public session of psychotherapy.[7] However, Popnedeleva's devotion to the intersectional understanding of equality is strengthened by her postsocialist experience and social situatedness. While the development of intersectional theory in the West was fueled by debates about race, class, and sexuality, as well as identity politics, postsocialist intersectionality stems from the inter-relations between a Socialist past, with the experience of totalitarian and authoritarian political regimes, and the everyday encounters with corruption, bureaucracy, nepotism, and political impunity as a continuing communist legacy, which has produced specific forms of postsocialist precarity (Suchland 2021), as well as corporeal and affective vulnerabilities. It might seem paradoxical to label Popnedeleva's work as a "soft" feminist position when its intrinsic signature is a critique of imperial powers. Likewise, its orientation towards other forms of inequality, makes it seem perhaps more inclusive, while softness and other "traditional" feminine traits enact a politics of strategical essentialism that promotes the reversal of binary hierarchies and encourages critical interrogations, in order to affirm the power of the "weak sex" and highlight its strategies of everyday resistance.

Perhaps all of these strategies have to be thought in relation to the resistance of the weak or, to put it in the words of philosopher Ewa Majewska, the "weak resistance" that is far from heroic and with hybrid ethical qualities, troublesome gender assumptions, and unholy origins. It may consist of gestures of weakness and fear and apparently meaningless private acts of refusal (Majewska 2016, p. 16) and while undermining the hegemony

of heroic agency and avoiding "easy fix" types of solutions, minoritarian positions and perspectives are prioritized (Majewska 2021, p. 153). From the affective point of view, "weak resistance" nourishes solidarity and shared empathy, sustaining and supporting each other, as well as persisting throughout long-term oppression (Majewska 2021, p. 131). What seems most important is that the purpose of weak resistance is not victory, but everyday survival (Majewska 2021, p. 139). The feminisms gathered by this article are attuned to this modality—they are missing spectacular heroism and theoretical virtuosity, but instead are willing to embrace ambiguity and troublemaking, despite the ongoing official rhetoric of the achieved gender quality and post-feminist sentiments. These feminisms share traces of "uneventfulness"—a characteristic mark suggested by Ukrainian scholars Mayerchyk and Plakhotnik (2021)—due to their informality and non-engagement with public institutions. The obscure patterns of postsocialist feminisms privilege intimacy, anonymity, low visibility, and unsuccessfulness, and rather focus on the repetitive patterns of everyday life and its bodily and affective manifestations instead of historical raptures, grand narratives, and turning points.

### Visceral Aesthetics: Getting under One's Skin

While the feminist designations discussed initially referred to an individual artistic or research position, and their authors used them rather as adjectives or epithets, they also present a critical vocabulary whose political potential I have tried to reveal. The orientation of these feminisms towards a productive opacity, critiques of established geopolitical patterns of visibility, as well as an investment in one's personal experience and everyday life, advance the categories of how to understand feminisms with intimacy and darkness as their central motives. A closer look at them may provide additional points of reference, enabling a more solid framing of "feminisms of their own", as a means to mirror the invisibility, opacity, and discursive elusiveness of feminist insights and sensibilities in Central and Eastern European art, as well as provide new theoretical tools to reflect upon their marginal, fragmented, and inconsistent condition. Since intimacy may be facilitated by darkness, I will start with the last concept.

Although in the interview with Červinkova and Šaldova, the artist alludes to the situation " . . . when it's dark", I suggest understanding darkness as a partial, not total inhibition of vision. Therefore, it can be submerged into the already existing body of research on opacity, but with a postsocialist feminist touch. Postcolonial scholars, such as Homi Bhabha, Gayatri Chakravorty Spivak, and Édouard Glissant among others, have explored the condition of opacity as a cultural strategy of resistance (Bhabha 1985; Spivak 1988; Glissant 1997). It provides a mechanism that prevents the transformation of subjects into classifiable and predictable objects of knowledge. Although feminist theories often appreciate visibility, this also entails the risk of being subjected to stereotypes, violence, and epistemic inequality, and a perspective of being explained according to the leading, often generalizing paradigms of knowledge, inattentive to details and differences.

In the postsocialist feminist context, opacity has been considered recently, for example, by scholar Raili Marling who argues that "visible and publicity-friendly feminism may be submerged in the broader neoliberal rationality and lose its ability to meaningfully challenge hierarchies of power" (Marling 2021, p. 95). Opacity and non-transparency not only resist panoptic neoliberalism—one of the key twists of the postsocialist condition—but also challenge post-imperial colonizing relations, in which the specific features and peculiarities of marginal territories are erased, but the assessment of artistic processes become pigeon-holed and explained within the framework of "Western feminism" and "Western knowledge". These considerations are especially valuable for postsocialist feminist discourses where a common problem is frustration, confusion, and a reluctance to embrace a visibly feminist identity. Postsocialist feminism rather manifests in a sensibility, in a feeling under one's skin, not a label, and welcomes theoretical cross-breeding and hybridity. Opacity resists the demand for a pure identity without contradictions and enables "living on the border" (Majewska 2021, p. 142).

One way to prevent any simplistic transparency about postsocialist feminism is to focus on local and often culturally peripheral phenomena and processes, acknowledging the complexity and internal contradictions. In the context of postsocialist feminism, this means building a feminist genealogy grounded in regional history, including a rethinking of the legacy of socialism. Opacity is a strategy that allows for the preservation of creativity, as well as for the possibility of surprise and the formation of new relationships. It helps to avoid integration in hierarchical relationships or dominant narratives, and views, based on stereotypical generalizations and clichéd designations. In this sense, being incomprehensible, unknowable, and invisible, is already a political position.

Likewise, opacity provides grounds for questioning the relationship between the center and the periphery. In Eastern European art history, these attempts have been undertaken, e.g., by art theoretician Piotr Piotrowski (2014) and further developed by subsequent art researchers who focus on the concepts of horizontal and alter-global art histories. These perspectives provoked paradigmatic changes in writing art history by counteracting a fundamental assumption that the cultural experiences of the Western world and their description, as well as the canon of selected masterpieces and a historical narrative built upon the succession of artistic styles, "can serve as a universal model, providing paths for "peripheries" to follow", as aptly summarized by Beáta Hock (2018, p. 3). Simply put, in darkness, it becomes more complicated to discern the center from the periphery, and the established categories and narratives might start to blur. Where the vision is impaired, other senses regain their importance, for example, touch, gut feelings, and visceral sensations. These new points of departure suggest an epistemic model of embodied knowledge and the lived experience—two strategies that are cherished by feminist epistemology, but often disregarded in art history. By registering affective and bodily manifestations, visceral aesthetics introduces the notion of intimacy which also underpins the relationships of empathy and solidarity, a way of mutual support, care, and help, touching each other and sharing the world in order to survive.

The embodied, affective and psychic aspects of intimate entanglements generate a particular feminist sensibility that can have different names (intuitive, reluctant, etc.) and is not only "close to one's skin" or even "gets under it". These turns of phrases, borrowed from feminist affect theorist Sara Ahmed (2010, 2017), remind us of the visceral dimension within feminism itself, its capacity, not only to touch and to envelop a body, but to soak in it. These soakings move the subjects beyond fixed and stable subject positions, the notion of sovereignty and self-masterhood, towards the transformation and nomadic movement (Braidotti 1994) that traverse the patterns of everyday life with its corporeal and affective experiences. Perhaps, this might be the reason why postsocialist feminisms do not stick to the "Anglo-American axis", but claim positions by its side, beyond it, and away from it, as "Close Others" in intimate spaces and moving across borders.

The traces of "other feminisms" scattered in the art history and theory of the region should not be considered as seemingly insignificant formations coming from a marginalized territory, nor as local variants of "true" feminism (which do not require clarification), but as a critical potentiality for re-evaluating the geographies of the center and periphery, the temporalities of "before" and "after", and the binary hierarchies of original and copy. They present a critical vocabulary that encourages questioning the "geographically neutral" standard of feminism, reinforcing horizontal, non-hierarchical, and margin-oriented methods in feminist art history, reviewing the understanding of feminist art as a linear progression towards ever-increasing progress, as well as expanding transnational geographies and temporalities of feminist art and art theory and ramifying transnational feminist debates.

These "islands of feminist interest" can be perceived as a rhizomatic structure whose aspects of proximity and distance, familiarity, and strangeness are not absolute, but flexible. It remains open to the experience of the uncanny and the encounter with the Close Other that arranges feminist art from different regions and periods in a new kinship that is often intimate and dark. These arrangements use strategically traditional features of femininity (such as emotionality, softness, relationality, corporeality, and intuition) to

explore their critical aspects, as well as advance aesthetic categories indicating failure, opacity, inefficiency, uneventfulness, vulnerability, weakness, and viscerality, derived from unofficial everyday experiences and intimate encounters.

Feminist sensibilities of postsocialism enhance visceral aesthetics by focusing on a complex set of relationships that exist between ideology and corporeality, intuitive embodied experiences, and political sensibilities. While pronouncing the relationship of postsocialist feminism to contemporary feminist movements, they underscore the difference and ambiguous otherness which is at the same time close and uncanny, similar and not similar enough. These theoretical movements offer a novel perspective to rethink feminism in a postsocialist condition, e.g., they can suggest a solution to the problem of how to speak about feminism in late socialist and postsocialist art and how to negotiate the public denial or confusion about feminism in the arts of this period. Bringing these "islands of feminist interest" together into a whole, one can discover, not the home or the holy origin, but perhaps a trail where the dutiful daughters of postsocialist feminism might meet or re-imagine their mothers.

**Funding:** This research was funded by the Latvian State Culture Capital Foundation.

**Data Availability Statement:** Data sharing is not applicable.

**Conflicts of Interest:** The authors declare no conflict of interest.

## Notes

1    One can, of course, object to the overuse of Woolf's metaphor. However, as I have argued elsewhere, the feminist temporality in Latvia (and probably in other postsocialist countries as well) does not share the linear patterns of the established four-waves feminism model. Instead, Julia Kristeva's account of women's time could be more accurate, since it enhances returns and repetitions, as well as allows one to "get stuck in time", subsequently, in "The Room of One's Own" (Ventrella 2017; Kukaine 2021).

2    See, Koobak and Marling (2014); Marling and Koobak (2017); Marling (2021) to mention a few interrogations whose theoretical span also covers the Baltic states.

3    This question mimics Spivak's renowned "Can the subaltern speak?" (1988), pushing its intrinsic colonial critique to an absurd extreme.

4    Needless to say, Russia's war in Ukraine (from 2022-ongoing) has only exaggerated this desire, for example, by provoking a mass deconstruction of Soviet monuments and limiting the learning of the Russian language in schools.

5    It is important to mention, that although I link these terms to particular authors, this does not exclude a wider and more complicated genealogy.

6    ZDZ is an abbreviation of her name used both by the artist and consequently also by Alise Tīfentāle.

7    Although these works of Popdeneleva are produced in the last decades, I view her in the context of late socialism and postsocialist art since the artist was born in 1956.

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
