# Peer review of "Intimacy and Darkness: Feminist Sensibility in (Post)socialist Art"

_arts_

Round 1
Reviewer 1 Report
This is an excellent, clear overview of the different strands of postsocialist feminist sensibilities and art. I enjoyed reading it. There are some fantastic synthesizing statements (eg. “Intuitive feminism is idiosyncratic – it is closed, peculiar, and theoretically unbounded by established patterns of feminist inquiry. It rather offers a personal gesture of protest, than a collective revolt, and being untraceable and nomadic, it resists unhomeliness by finding a home in one’s skin.”). The research is comprehensive and the conclusions are convincing and thought-provoking. The only section I question is the one on parafeminism. Based on my understanding of Amelia Jones' parafeminism and its gender-fluid, non-binary orientation, I wonder if ZDZ’s self-portraits are a better example than the A House Near a River series. The self-portraits that are reproduced in Tifentale’s online article are wonderfully ambiguous and unconventional. If the current author wants to retain the River series as an example, I would suggest they state it more strongly that Tifentale does not really engage Jones’ concept as it was originally intended, and that the current author is using the label solely as a spatial metaphor (which is mentioned, but needs to be clarified further). Otherwise this article needs (very) minor revision for English language usage – eg. mostly around the use or absence of definite articles.
Author Response
Response to Reviewer 1
Point 1: This is an excellent, clear overview of the different strands of postsocialist feminist sensibilities and art. I enjoyed reading it. There are some fantastic synthesizing statements (eg. “Intuitive feminism is idiosyncratic – it is closed, peculiar, and theoretically unbounded by established patterns of feminist inquiry. It rather offers a personal gesture of protest, than a collective revolt, and being untraceable and nomadic, it resists unhomeliness by finding a home in one’s skin.”). The research is comprehensive and the conclusions are convincing and thought-provoking.
Response 1: Thank you very much for your kind words!
Point 2: The only section I question is the one on parafeminism. Based on my understanding of Amelia Jones' parafeminism and its gender-fluid, non-binary orientation, I wonder if ZDZ’s self-portraits are a better example than the A House Near a River series. The self-portraits that are reproduced in Tifentale’s online article are wonderfully ambiguous and unconventional. If the current author wants to retain the River series as an example, I would suggest they state it more strongly that Tifentale does not really engage Jones’ concept as it was originally intended, and that the current author is using the label solely as a spatial metaphor (which is mentioned, but needs to be clarified further).
Response 2: Thank you for your suggestions! I have included a reference also to the self-portraits and emphasized that Tīfentāle’s usage of the term is quite different from Jones’.
Point 3: Otherwise this article needs (very) minor revision for English language usage – eg. mostly around the use or absence of definite articles.
Response 3: I have tried to deal with it.
Reviewer 2 Report
The scholarly method is thorough and coherent, as far as it goes, and the range of reference is impressive, though partial. A similar assessment applies to the argument: it’s promising; but it stops short.
The quality of the prose is high. In some ways, the article reads like an introductory survey at the beginning of a book, whetting the appetite for what is to follow. (Worth thinking about?).
However, as an article, there are major issues with the writer’s treatment of the central concepts under discussion. There are two posssible ways of dealing with this.
The first is either to minimise unsupported references to insufficiently differentiated Anglophone feminisms, or to research them further, and then to clarify what the ideas of intimacy and darkness can contribute transnationally.
For example, the reference to US male art historian, Mark Allen Svede, who is insignificant in this scenario, is baffling, especially given the omission of many/most pre-eminent feminists (see below). He is cited as saying (20 years ago) that “women were not particularly disadvantaged in the institutional art world.” This is not the view of any feminist, or even prominent or serious generalist, writer on art in the UK or the US. It should be removed.
The second way of dealing with the overall issues goes deeper and could either replace the above approach, or be outlined towards the end as an outline for further study.
I hope that what follows doesn’t sound overly negative. In fact, I very much hope the author will be prepared to do the necessary and I encourage them to rebalance their work so that it might have the impact it deserves. There are some important issues here, but they remain insufficiently clear; worse, they risk being undermined by being published in too provisional a form.
Given that the aspiration is to expand ‘temporal geographies of transnational feminist debates’, a fuller exploration of transnational thought is required. The same is true of the postsocialist. There is also too little direct exploration of actual art works or practices.
First and foremost: Marsha Meskimmon’s work gets only a mention from 2007, whereas her 2020 book Transnational Feminisms, Transversal Politics and Art. Entanglements and Intersections is precisely in in the ballpark. What’s more, it is the first volume in her planned ambitious trilogy: v.2 Feminisms and Art’s Transhemispheric Histories: Ecologies and Genealogies; Transnational Feminisms and v 3 Posthuman Aesthetics: Resonance and Riffing.*
While acknowledging that the emphasis of the article under review is explicitly on the ‘postsocialist’, it is implicitly assumed that this term could only apply to Eastern European art and thought. Yet one could readily argue that possibly some US, and certainly Western European feminism, (and radical thought in general) has been struggling with the vacuum created by the defeat of the Left. Isn’t this an aspect of postsocialism? Whatever the answer, it needs exploring/analysing.
Gayatri Chakravorti Spivak similarly gets only a passing mention, whereas her 1988 elaboration of the ‘subaltern’ is highly relevant and highly influential.
Meanwhile other luminaries are left out altogether. Griselda Pollock, for example, has dominated feminist art history in the UK since the 1970s, and her ambitious intellectual frameworks are well worth discussing as postsocialist; eg, they explicitly and prominently include nuanced explorations of Marxism (eg in Vision and Difference, in which the second chapter is “Vision, Voice and Power: feminist art histories and Marxism”).
Jacqueline Rose’s books have much to offer any exploration of intimacy and darkness in art or indeed in general (eg Sexuality in the field of vision (2005); Mothers: An essay on love and cruelty (2018); and On Violence and On Violence Against Women (2021).
Bulgarian-French Julia Kristeva is a surprising omission (Black Sun (1980), Powers of Horror (1987?), to mention but 2 books).
I could go on (Elizabeth Grosz, Rosi Braidotti, Luce Irigaray, material feminists … very recent work by Black women artists internationally, and their belated but increasing recognition in London, notably Sonia Boyce and Turner prizewinner Lubaina Himid … Karen Barad’s scientific understanding of agency as enactment … etc, etc).
So I recommend the article for publication with significant revisions. The range of reference requires extension; and the argument needs to be more conclusive. The latter doesn't mean making a choice between binaries, but rather a fuller exploration of what the core concepts might reveal in each other (ie following through the dynamic of non-binary thought.
Closer analysis and perhaps comparison of a couple of specific artists whose work might be understood as transnational should be brought to bear on the notions of darkness and intimacy, which remain unspecified. The absence of psychoanalysis seems strange in relation to these concepts, if only to demonstrate their irrelevance. But I must say, feminist readings of Freud’s Uncanny would seem indispensable.
Author Response
Dear reviewer,
Thank you very much for your extensive review, and the time and energy you devoted to writing it! I appreciate your input.
I am happy to discover that, according to what you have written, the article makes multiple (potential) connections to feminist thinking traditions and such figures as Gayatri Chakravorti Spivak, Griselda Pollock, Julia Kristeva, Elizabeth Grosz, Rosi Braidotti, Luce Irigaray, etc. I interpret it as a signal that my perspective can find its own trajectory in the feminist world-making.
However, I doubt that bringing these authors straight into the article will make my argument more eloquent. In fact, I fear the opposite might happen. I strongly feel I don’t need to introduce any more references to already established feminist theories than it is now. In the article, I give voice to what has so far not been heard, and I need to ensure there is a proper acoustic space for that.
Nonetheless, to pay tribute to your comments and suggestions, I have emphasized the positionality of the article which can hopefully work as a disclaimer: I do indeed use the term ‘postsocialist’ in a way that does not immediately apply to ‘US, and certainly Western European feminism’. I feel that this usage of the term might be detrimental to Eastern and Central Europe by assimilating its cultural and political difference and again shifting the attention “to the West”.
I have added a few paragraphs to enhance “closer analysis and perhaps a comparison of a couple of specific artists”, as you have suggested, however, the length of the article does not allow me to go much deeper into that. Also, the article’s primary goal is to (re)define terms and revise vocabulary, instead of demonstrating how to apply them (that could be done somewhere else), and to provide some theoretical tools for discussing feminist questions in regional contexts and overcoming the popular assumption of its theoretical “void”.
As for your suggestion concerning Mark Allen Svede, “who is insignificant in this scenario” because his views are not shared by “any feminist, or even prominent or serious generalist, writer on art in the UK or the US”, I hope you understand that it reads almost like an anecdote. Please let me judge the significance of the quoted authors, and Svede is an important figure in the context.
All best wishes,
The author